# Improved Remote Sensing Image Classification Based on Multi-Scale Feature Fusion

**Chengming Zhang** [1,2,3,*,†] **, Yan Chen** [1,3] **, Xiaoxia Yang** [1,†] **, Shuai Gao** [4] **, Feng Li** [5] **, Ailing Kong** [1] **, Dawei Zu** [1] **and Li Sun** [1]

1 College of Information Science and Engineering, Shandong Agricultural University, 61 Daizong Road, Taian 271000, China
2 Key Open Laboratory of Arid Climate Change and Disaster Reduction of CMA, 2070 Donggangdong Road, Lanzhou 730020, China
3 Shandong Technology and Engineering Center for Digital Agriculture, 61 Daizong Road, Taian 271000, China
4 Chinese Academy of Sciences, Institute of Remote Sensing and Digital Earth, 9 Dengzhuangnan Road, Beijing 100094, China
5 Shandong Provincal Climate Center, No. 12 Wuying Mountain Road, Jinan 250001, China
* Correspondence: chming@sdau.edu.cn; Tel.: +86-139-5382-3659
† These authors are co-first authors as they contributed equally to this work.

**Abstract:** When extracting land-use information from remote sensing imagery using image segmentation, obtaining fine edges for extracted objects is a key problem that is yet to be solved. In this study, we developed a new weight feature value convolutional neural network (WFCNN) to perform fine remote sensing image segmentation and extract improved land-use information from remote sensing imagery. The WFCNN includes one encoder and one classifier. The encoder obtains a set of spectral features and five levels of semantic features. It uses the linear fusion method to hierarchically fuse the semantic features, employs an adjustment layer to optimize every level of fused features to ensure the stability of the pixel features, and combines the fused semantic and spectral features to form a feature graph. The classifier then uses a Softmax model to perform pixel-by-pixel classification. The WFCNN was trained using a stochastic gradient descent algorithm; the former and two variants were subject to experimental testing based on Gaofen 6 images and aerial images that compared them with the commonly used SegNet, U-NET, and RefineNet models. The accuracy, precision, recall, and F1-Score of the WFCNN were higher than those of the other models, indicating certain advantages in pixel-by-pixel segmentation. The results clearly show that the WFCNN can improve the accuracy and automation level of large-scale land-use mapping and the extraction of other information using remote sensing imagery.

**Keywords:** convolutional neural network; image segmentation; multi-scale feature fusion; semantic features; Gaofen 6; aerial images; land-use; Tai'an

## 1. Introduction

Remote sensing images have become the main data source for obtaining land-use information at broad spatial scales. The most common method first assigns each pixel to a category using image segmentation and subsequently generates land-use information according to the pixel-by-pixel classification result [1,2]. Since the accuracy of the final extraction is determined by the accuracy of the pixel-by-pixel classification, improving the segmentation accuracy is a common research focus [3]. The pixel feature extraction method and the classifier performance both have a decisive influence on segmentation results [4,5].

Researchers have proposed a variety of methods to extract ideal features. For example, spectral indexes have been widely applied in the classification of low- and medium-spatial-resolution remote sensing images, as they can accurately reflect statistical information regarding pixel spectral values. Commonly used indexes include the vegetation index [6–13], water index [14], normalized difference building index [15], ecological index [16], normalized difference vegetation index (NDVI) [10,13], and derivative indexes such as the re-normalized difference vegetation index [12] or the growing season normalized difference vegetation index [13]. Taking account of the advantage of the short time period of low- and medium-spatial-resolution remote sensing images, some researchers have applied spectral indexes to time-series images [17,18]. However, such indexes are mainly used to express common information within the same land-use type through simple band calculations. When these are applied to remote sensing images with rich detail and high spatial resolution, it becomes difficult to extract features with good discrimination, limiting the application of spectral indexes to the classification of such images.

For high-spatial-resolution remote sensing images with a high level of detail, researchers initially proposed the use of texture features with the gray matrix method [19]. Subsequently, other researchers have proposed a series of methods for extracting more abundant texture features, including the Gabor filter [20], the Markov random field model [21], the Gibbs random field model [22], and the wavelet transform [23]. Compared with spectral index features, texture features can better express the spatial correlation between pixels, improve the ability to distinguish between features, and effectively improve the accuracy of pixel classification.

Although the combination of spectral and texture features has greatly promoted the development of remote sensing image segmentation technology and significantly improved the accuracy of pixel classification results, the ongoing improvement in remote sensing technology has resulted in increasing image resolutions and higher levels of detail. Traditional texture feature extraction techniques now struggle with high-resolution images, such that new methods are needed to obtain effective feature information from such images [24,25].

The developing field of machine learning is currently being applied to pixel feature extraction, with early applications to image processing, including neural networks [26,27], support vector machines [28,29], decision trees [30,31], and random forests [32,33]. These methods use pixel spectral information as inputs and achieve the desired feature results through complex calculations. Although these methods can fully explore the relationship between channels and obtain some effective feature information, these features only express the information of a single pixel rather than the spatial relationship between pixels, limiting these methods' application in the processing of remote sensing images.

Convolutional neural networks (CNNs) use pixel blocks as inputs and compute them with a set of convolution kernels to obtain more features with a stronger distinguishing ability [34–37]. The biggest advantage of CNNs lies in their ability to simultaneously extract specific pixel features as well as spatial features of pixel blocks. This approach can reasonably set the structure of the convolutional layer according to the characteristics of the image and the extraction target, so as to extract features that meet the requirements, achieving good results in image processing [34,36]. The most widely used CNNs in the field of camera image processing include fully convolutional networks (FCNs) [38], SegNet [39], DeepLab [40], RefineNet [41], and U-NET [42]. DeepLab expands the convolution for rich features in camera images, enabling the model to effectively expand the receptive field without increasing the calculations required. SegNet and UNET can establish a symmetrical network structure and segment camera images, resulting in an efficient utilization of high-level semantic features. RefineNet uses a multipath structure to combine coarse high-level semantic features and relatively fine low-level semantic features by equal-weight fusion.

As CNNs have outstanding advantages in feature extraction, they have been widely used in other fields, such as real-time traffic sign recognition [43], pedestrian recognition [44], apple target

recognition [45], plant disease detection [46], and pest monitoring [47]. Researchers have established a method to extract coordinate information of an object from street view imagery [48] using CNNs.

Compared to camera images, remote sensing images have fewer details and more mixed pixels. When using a CNN to extract information from remote sensing images, the influence of the convolution structure on feature extraction must be considered [49]. Existing CNN structures are mainly designed for camera images with a high level of detail; therefore, a more ideal result can be obtained through adjustments that consider the specific characteristics of remote sensing images [50,51]. Researchers have proposed a series of such adjustments for applying CNNs to remote sensing image processing [3] and some classic CNNs have been widely applied in this field [52,53]. Based on the characteristic analysis of target objects and specific remote sensing imagery, researchers have established a series of CNNs such as two-branch CNN [49], WFS-NET [54], patch-based CNN [55], and hybrid MLP-CNN [56]. Researchers have also used remote sensing images to create many benchmark datasets, such as EuroSAT [57] or the Inria Aerial Image dataset [58], to test the performance of CNNs.

In this study, we established a CNN structure based on variable weight fusion, the weight feature value convolutional neural network (WFCNN), and experimentally assessed its performance. The main contributions of this work are as follows.

- Based on the analysis of the data characteristics of remote sensing images, fully considering the impact of image spatial resolution on feature extraction, we establish a suitable network convolution structure;
- The proposed approach can effectively fuse low-level semantic features with high-level semantic features and fully considers the data characteristics of adjacent areas around objects on remote sensing images. Compared with the strategies adopted by other models, our approach is more conducive to effective features.

## 2. Datasets

We employed two sets of datasets to test the performance of the model. We created the GF-6 images dataset that contains 587 image-label pairs. The aerial image labeling dataset was the benchmark dataset [58].

### *2.1. The GF-6 Images Dataset*

#### 2.1.1. Study Area

Tai'an is a prefecture-level city covering ~7761 km$^2$ in Shandong Province, eastern China (116°20–117°59′E, 35°38′–36°28′N; Figure 1); it has jurisdiction over the districts of Taishan and Daiyue, the cities of Feicheng and Xintai, and the counties of Ningyang and Dongping. Its terrain is highly variable, including mountains, hills, plains, basins, and lakes (Figure 1c). The mountains are concentrated in the east and north, including Mount Tai (one of the Five Great Mountains of China). The hills are mainly distributed in southwestern Xintai, eastern Ningyang, the northwestern city suburbs, southern Feicheng, and northern Dongping. The major basin lies within Dongping and contains Dongping Lake, Daohu Lake, and Zhoucheng Lake.

In the study area, crops are divided into summer and autumn crops, according to the growing season. Summer crops mainly refers to winter wheat, and its growth period begins in the autumn and reaches the early summer of the second year. Autumn crops mainly refer to corn, millet, and potato. The growth period generally ranges from early summer to early winter. Therefore, crop planting areas are usually divided into winter wheat and farmland.

There are eight main land-use types in the study area: developed land, water bodies, agricultural buildings, roads, farmland, winter wheat, woodland, and others. Developed land includes residential and factory areas, agricultural buildings refer to buildings in crop planting areas, and others refer to areas not used. These diverse landforms and land-uses render the study area representative of many different regions and suitable as an experimental area for this study.

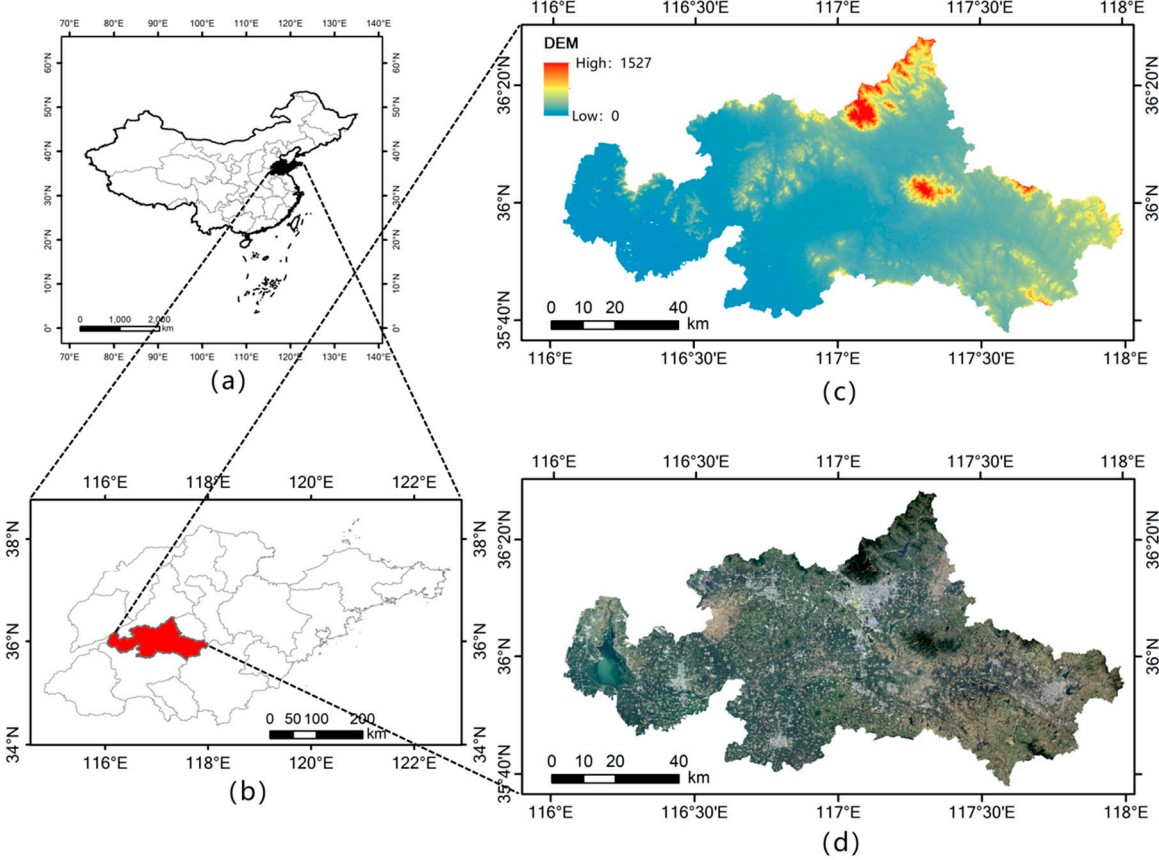

**Figure 1.** Geographic location of the city of Tai'an (red boundary) within Shandong Province, China: (**a**) Geographic location of China; (**b**) Geographic location of Shandong Province; (**c**) Terrain of Tai'an and (**d**) remote sensing images used in this study.

### 2.1.2. Remote Sensing Data

Gaofen 6 (GF-6) is a low-orbit optical remote sensing satellite launched by China in 2018, which provides images with high resolution, wide coverage, high quality, and high efficiency. GF-6 has a design life of eight years and two cameras: a full-color 8 m (high-resolution) multi-spectral camera with an image swath of 90 km and a 16 m (medium-resolution) multi-spectral wide-format camera with an image swath of 800 km. The GF-6 satellite will operate with the Gaofen 1 satellite network, reducing the time resolution of remote sensing data acquisition from 4 d to 2 d. In-depth analysis of these remote sensing images and the establishment of appropriate image segmentation methods will allow the acquisition of high-precision global land-use information, which is of great significance for improving the application level of GF-6.

We selected images from different seasons to increase the anti-interference abilities of the WFCNN and mitigate potential complications, such as the change in seasons, and thus enhance applicability. We collected a total of fifty-two GF-6 remote sensing images (Figure 1d). These images were divided into four groups according to the image acquisition time. The first image group was captured in autumn 2018, the second in winter 2018, the third in spring 2019, and the fourth in summer 2019. Together, the images of each group covered the study area. Specifications are given in Table 1.

**Table 1.** Specifications for Gaofen 6 satellite imagery used in this study.

| Band | Range (µm) | Spatial Resolution (m) | Width (km) |
|---|---|---|---|
| Panchromatic | 0.45–0.90 | 2 | >90 |
| B1 | 0.45–0.52 | 8 | >90 |
| B2 | 0.52–0.60 | 8 | >90 |
| B3 | 0.43–0.69 | 8 | >90 |
| B4 | 0.76–0.90 | 8 | >90 |

Image preprocessing included geometric correction, radiation correction, and image fusion. We used Python to develop a program for geometry correction that captured control points from geometrically corrected Gaofen 2 (GF-2) remote sensing images. These geographically corrected GF-2 images have a spatial resolution of 1 meter, which is suitable for selecting control points from them.

Atmospheric correction was performed using the fast line-of-sight atmospheric analysis of spectral hypercubes module in the environment for visualizing images (ENVI) software. The multi-spectral and panchromatic images were fused using the Pan-sharping module in ENVI. The resulting image included four bands (blue, green, red, and near-infrared) with a 2-m spatial resolution.

### 2.1.3. Dataset Creation

In the images captured in winter or spring, developed land, water bodies, agricultural buildings, roads, bare fields, and farmland can be directly distinguished by visual interpretation within ENVI. To accurately distinguish winter wheat and woodland, 353 sample points were obtained through ground surveys (118 woodland and 235 winter wheat, Figure 2). Woodland areas had a rough texture, a large color change, and an irregular shape, while winter wheat areas were finer, smoother, and generally regular in shape. Defining these features helped to improve the accuracy of visual interpretation.

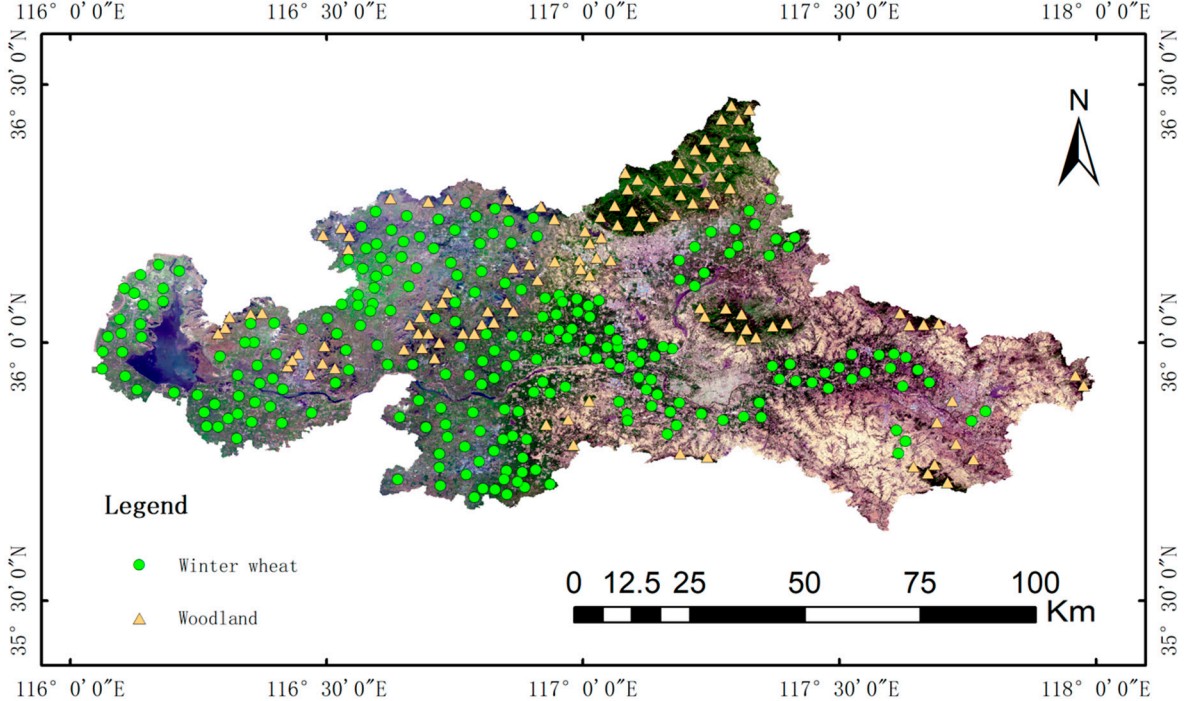

**Figure 2.** Geographic location of ground survey sample points used to distinguish winter wheat (green) and woodland (yellow).

The ENVI software was used to fuse the images into a complete mosaic, covering the whole study area. This image was then segmented into patches of 480 × 360 pixels, from which 587 images were

selected for manual classification based on the eight land-use types defined above. Numerical codes were then assigned to these land-use types; an example is provided in Figure 3. All labeled image patches and their labeled files formed data sets for training and testing the WFCNN model.

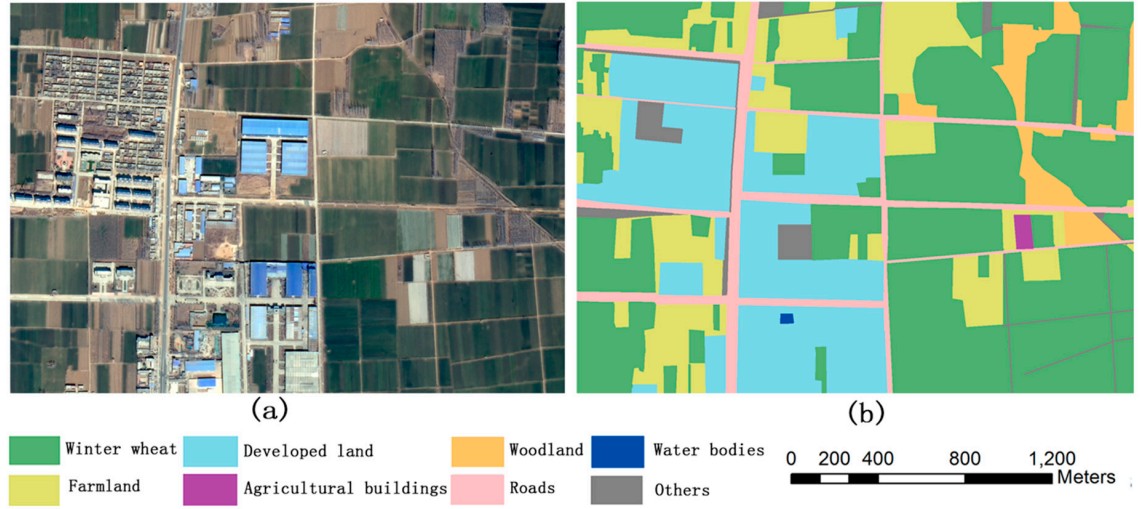

**Figure 3.** Example of land-use classification: (**a**) original image and (**b**) classified image.

## 2.2. The Aerial Image Labeling Dataset

The aerial image labeling dataset was downloaded from https://project.inria.fr/aerialimagelabeling/ [58]. The images cover dissimilar urban settlements, ranging from densely populated areas (e.g., the financial district of San Francisco) to alpine towns (e.g., Lienz in the Austrian Tyrol). The dataset features were as follows:

Coverage of 810 km$^2$
Aerial orthorectified color imagery with a spatial resolution of 0.3 m
Ground truth data for two semantic classes: building and no building

The original aerial image labeling dataset contained 180 color image tiles, 5000 × 5000 pixels in size, which we cropped into small image patches, each with a size of 500 × 500 pixels. An image-label pair example is provided in Figure 4.

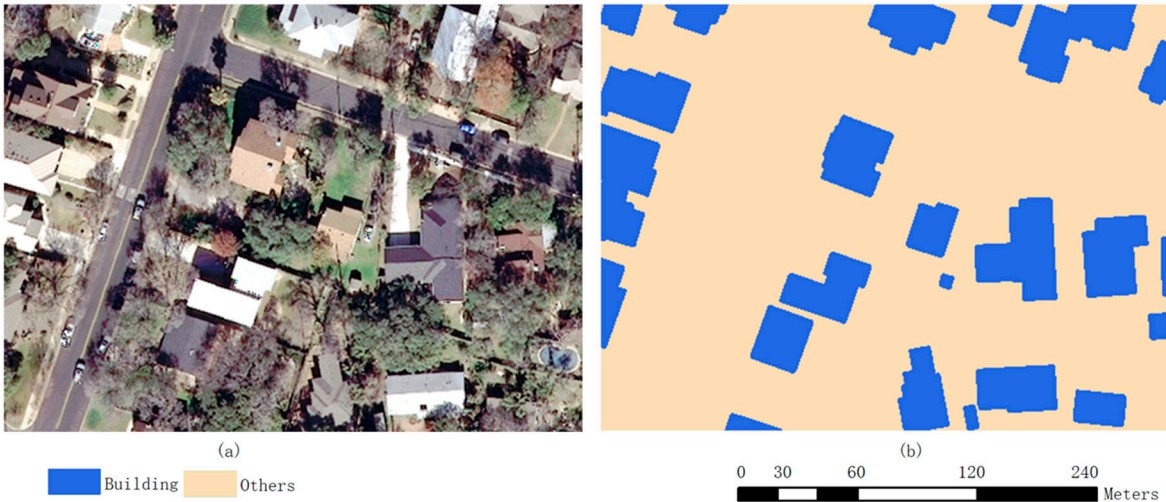

**Figure 4.** Example of image-label pair: (**a**) original image and (**b**) classified image.

## 3. Methods

### 3.1. Structure of the WFCNN Model

The WFCNN model includes an encoder, a decoder, and a classifier (Figure 5). The encoder is used to extract pixel-by-pixel features, while the decoder is used to fuse the coarse high-level semantic features and fine low-level semantic features. The classifier is used to complete the pixel-by-pixel classification. When training the model, the image patch and the corresponding label file are used as inputs. Once the model is successfully trained, the image to be segmented is used as the input, with the output being pixel-by-pixel label files.

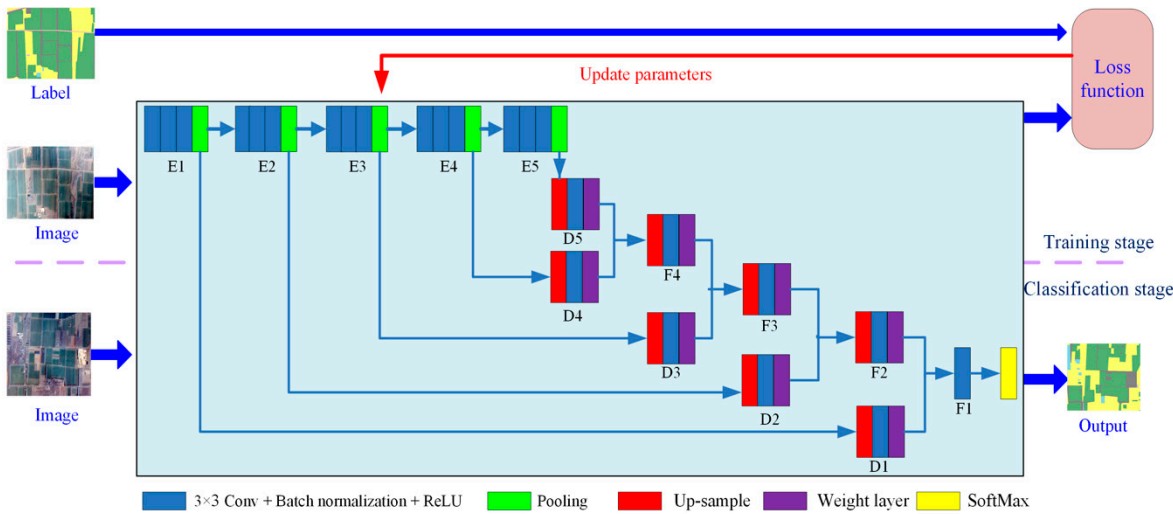

**Figure 5.** Structure of the weight feature value convolutional neural network model.

### 3.1.1. Encoder

The encoder consists of five serial connection feature extraction units that can extract five levels of semantic features for each pixel. Each unit consists of three convolutional layers, a batch normalization layer, an activation layer, and a pooling layer. The encoder contains a total of 15 convolutional layers with differing amounts of convolution kernels (Table 2). The advantage of this structural design is that the influence of the image's spatial resolution and geographical coverage is fully considered. Ensuring that sufficient semantic features can be extracted avoids the risk that the excessively deep convolution structure may cause noise in extracted feature values. This is beneficial to the classifier when performing pixel-by-pixel classification.

**Table 2.** Number of convolution kernels for each convolutional layer.

| Layer | Number of Convolution Kernels |
|---|---|
| 1, 2, 3 | 64 |
| 4, 5, 6, | 128 |
| 7, 8, 9 | 256 |
| 10, 11, 12, 13, 14, 15 | 512 |

The activation layer uses the widely used rectified linear unit function as the activation function. Commonly, the pooling operation can accelerate feature aggregation and eliminate feature values with poor discrimination, which is beneficial to the operation of the classifier. However, the pooling operation generally leads to a reduced feature map resolution, affecting segmentation accuracy. When pooling the edge area of two object types, the fact that two adjacent pixels often belong to different categories makes it is easy to mistakenly apply the feature values of one category to another category.

When the pooling operation is carried out on images with a high level of detail or in the shallow convolutional structure, the influence of this problem is not obvious, but if this operation is carried out in the deep convolutional structure, the problem has a serious influence on feature extraction.

WFCNN uses a new pooling strategy to solve this problem. In the E1, E2, and E3 feature extraction units, a $2 \times 2$ pool core is used and the pooling step length is 2, encouraging the advantages of the pooling operation in accelerating feature aggregation. In the later stages, due to the large difference between the current resolution and that of original images, the step length of the pooling layer in E4 and E5 cells is adjusted to 1. Here, the pool kernel size is still $2 \times 2$, but when the size of the function block is smaller than that of the kernel participating in the pool operation, the size of the kernel pool is adjusted to match the function block. This ensures that a valid pool value is obtained, so the size of the feature graph output through E4 and E5 remains the same.

### 3.1.2. Decoder

The decoder is composed of five decoding units (D5, D4, D3, D2, and D1) that correspond to and decode the encoding units with the same number (Figure 4). The fusion units F4, F3, F2, and F1 are used to fuse the feature values obtained by the decoding unit. Other than F1, which only contains one pooling layer, the structure of the other units is the same. Each unit includes one up-sampling layer, several convolution layers, and one weight layer.

The up-sampling layer is a deconvolution layer, which is used to restore the size of the feature graph. WFCNN adopts a gradual recovery strategy, in which the number of rows and columns are respectively doubled during each adjustment; the number of rows and columns of the feature graph are eventually restored to be consistent with the original image.

The convolutional layer adjusts the feature values after up-sampling to ensure the consistency of the structure of the feature graphs involved in fusion. The adjustment strategy adopted by WFCNN reduces the depth of the feature graph (Table 3). The decoder finally generates a feature vector consisting of 64 elements for each pixel.

**Table 3.** Number of feature layers per decoding unit.

| Unit | Layers before Adjustment | Layers after Adjustment |
|------|--------------------------|-------------------------|
| D5 | 512 | 512 |
| D4 | 512 | 512 |
| F4 | 512 | 256 |
| D3 | 256 | 256 |
| F3 | 256 | 128 |
| D2 | 128 | 128 |
| F2 | 128 | 64 |
| D1 | 64 | 64 |
| F1 | 64 | 64 |

Each weight layer contains only one convolution kernel of type $1 \times 1 \times h$, which is used to unify the feature values for stretching or narrowing transformation before fusion. The essence of the weight layer is to uniformly multiply a certain coefficient for a certain feature layer and convert it into a convolutional operation, to ensure that the model can be trained end-to-end.

### 3.1.3. Classifier

The Softmax model is a widely used classifier in FCN, SegNet, DeepLab, RefineNet, UNET, and other models. In WFCNN, Softmax uses the 64-layer feature graph generated by the decoder as input, calculates the probability of the pixel belonging to each category pixel-by-pixel, and organizes it into a category probability vector as the output. The WFCNN uses the category corresponding to the maximum probability value as the pixel category.

### 3.2. Training WFCNN

#### 3.2.1. Loss Function

WFCNN defines the loss function based on the cross entropy of the sample:

$$H(p, q) = -\sum_{i=1}^{m} q_i \log(p_i) \tag{1}$$

where $m$ is the number of categories, $p$ is the category probability vector with a length of $m$ output by WFCNN, and $q$ is the real probability distribution, generated according to the manual label. In $q$, the components are 0 except for that corresponding to the pixel's category, where it is 1. In the WFCNN, each pixel is regarded as an independent sample, allowing the loss function to be defined as:

$$loss = -\frac{1}{total} \sum \sum_{i=1}^{m} q_i \log(p_i) \tag{2}$$

where *total* represents the sample count.

#### 3.2.2. Training Algorithm

WFCNN is used end-to-end, with a stochastic gradient descent algorithm [50] used as the training algorithm with the following steps:

1.  The hyperparameters in the training process are determined and the parameters of the model are initialized.
2.  The selected image-label pairs are input into the model as training data.
3.  The model carries out a forward calculation on the current training data.
4.  Equation (2) is used to calculate the loss function of the real probability distribution and the predicted probability distribution.
5.  The random gradient descent algorithm is used to update the parameters of the model and complete the training process.
6.  Steps (3), (4), and (5) are repeated until the loss function is less than the specified expected value.

### 3.3. Experimental Setup

We used the SegNet, U-Net, and RefineNet models for comparison with WFCNN, as their structures and working principles are similar, providing the best test for WFCNN. The models were set up based on previous research, with SegNet containing 13 convolutional layers [39], U-Net containing 10 convolutional layers [42], and RefineNet containing 101 convolutional layers [41]. We implemented WFCNN based on the TensorFlow Framework, using the Python language. To better assess its performance, we also tested two variants, termed WFCNN-1 and WFCNN-2 (Table 4).

**Table 4.** Models used in the comparative experiment.

| Name | Description |
|---|---|
| WFCNN | |
| SegNet | Similar to WFCNN, the classifier used only high-level semantic features. |
| U-Net | Similar to WFCNN, the classifier used fused features. |
| RefineNet | Similar to WFCNN, the linear model was also adopted for feature fusion, but the parameters were all fixed as 1. |
| WFCNN-1 | The decoding unit was modified to use an adjustment strategy for the feature map depth ascending scale. The length of the feature vector generated by the decoder was 512 for each pixel. |
| WFCNN-2 | The decoding unit was modified to remove the weight layer. |

All experiments were conducted on a graphics workstation with a 12 GB NVIDIA graphics card and the Linux Ubuntu 16.04 operating system, using the data set defined in Section 2.

To increase the number and diversity of samples, each image in the training data set was processed using color adjustment, horizontal flip, and vertical flip steps. The color adjustment factors included brightness, saturation, hue, and contrast, and each image was processed 10 times. These enhanced images were only used as training data.

We used cross-validation for comparative experiments. When using the GF-6 images dataset, 157 images were randomly selected as test data every training round, and the other images were used as training data until all images were tested once. When using the aerial image labeling dataset, 3000 images were used as test data every training session.

## 4. Results

Overall, 10 results tested on the GF-6 images dataset were randomly selected from all models (Figure 6), and 10 results were tested on the aerial image labeling dataset (Figure 7). As can be seen from Figures 6 and 7, WFCNN performed best in all cases.

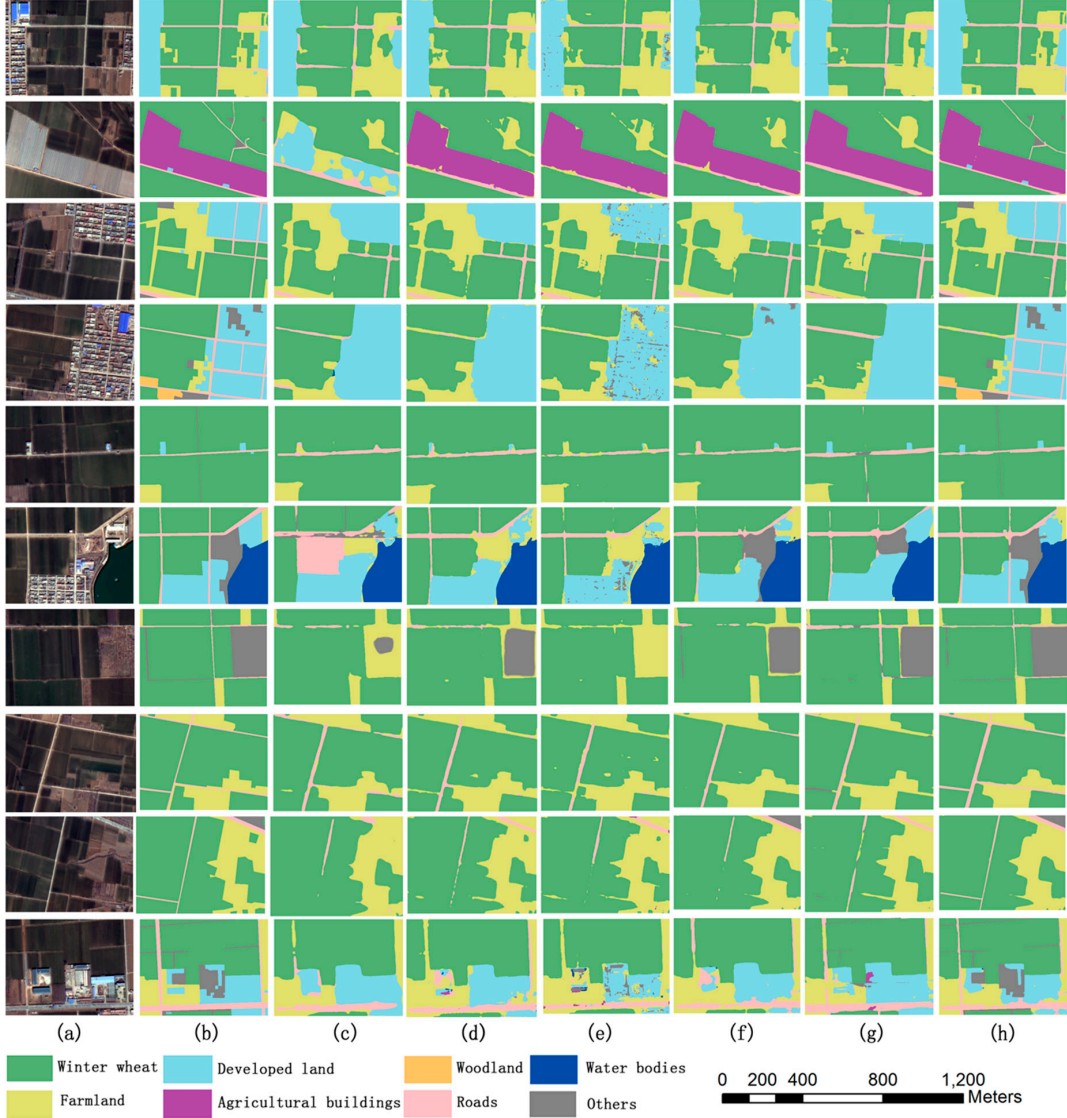

**Figure 6.** Comparative experimental results for 10 selected images from the GF-6 images dataset: (**a**) original GF-6 image, (**b**) manual classification, (**c**) SegNet, (**d**) U-Net, (**e**) RefineNet, (**f**) WFCNN-1, (**g**) WFCNN-2, and (**h**) WFCNN.

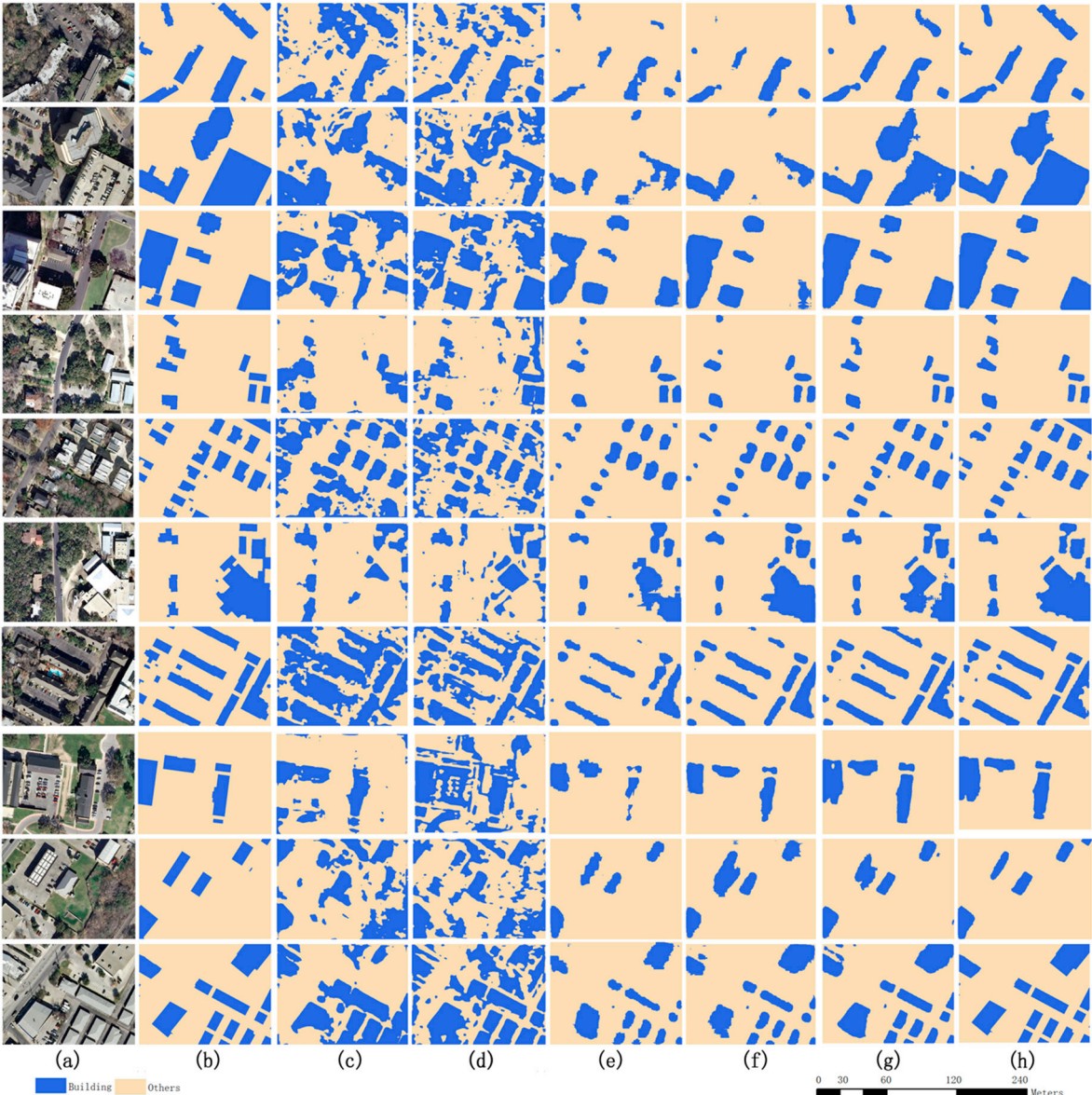

**Figure 7.** Comparative experimental results for 10 selected images from the aerial image labeling dataset: (**a**) original image, (**b**) manual classification, (**c**) SegNet, (**d**) U-Net, (**e**) RefineNet, (**f**) WFCNN-1, (**g**) WFCNN-2, and (**h**) WFCNN.

SegNet exhibited most errors and the distribution was relatively scattered, with more misclassified pixels in both the edge and inner areas. This shows that it is more reasonable to combine high semantic features with low semantic features than to use only high semantic features.

The number of misclassified pixels produced by all three variants of WFCNN was lower for RefineNet and U-net. The excellent performance of RefineNet and U-Net in the camera image indicates that the network structure should be determined in accordance with the spatial resolution of the image.

WFCNN performed better than WFCNN-1. This excellent performance indicates that the feature vector dimension was too high and is not conducive to improving the accuracy of the classifier. The result that WFCNN performed better than WFCNN-2 shows that the weight layer played a role.

By comparing the performance of U-Net, RefineNet, WFCNN-1, WFCNN-2, and WFCNN, it indicates that different feature fusion methods differ in their contributions to improving accuracy, so it is necessary to choose an appropriate feature fusion method for a given situation.

Figures 8 and 9 present confusion matrices for the different models, which again demonstrate that WFCNN had the best segmentation results. Comparing Figures 8 and 9, it can be found that the performance of each model on the aerial image labeling dataset is better than that on the GF-6 images dataset, indicating that the spatial resolution has a certain impact on the performance of the model.

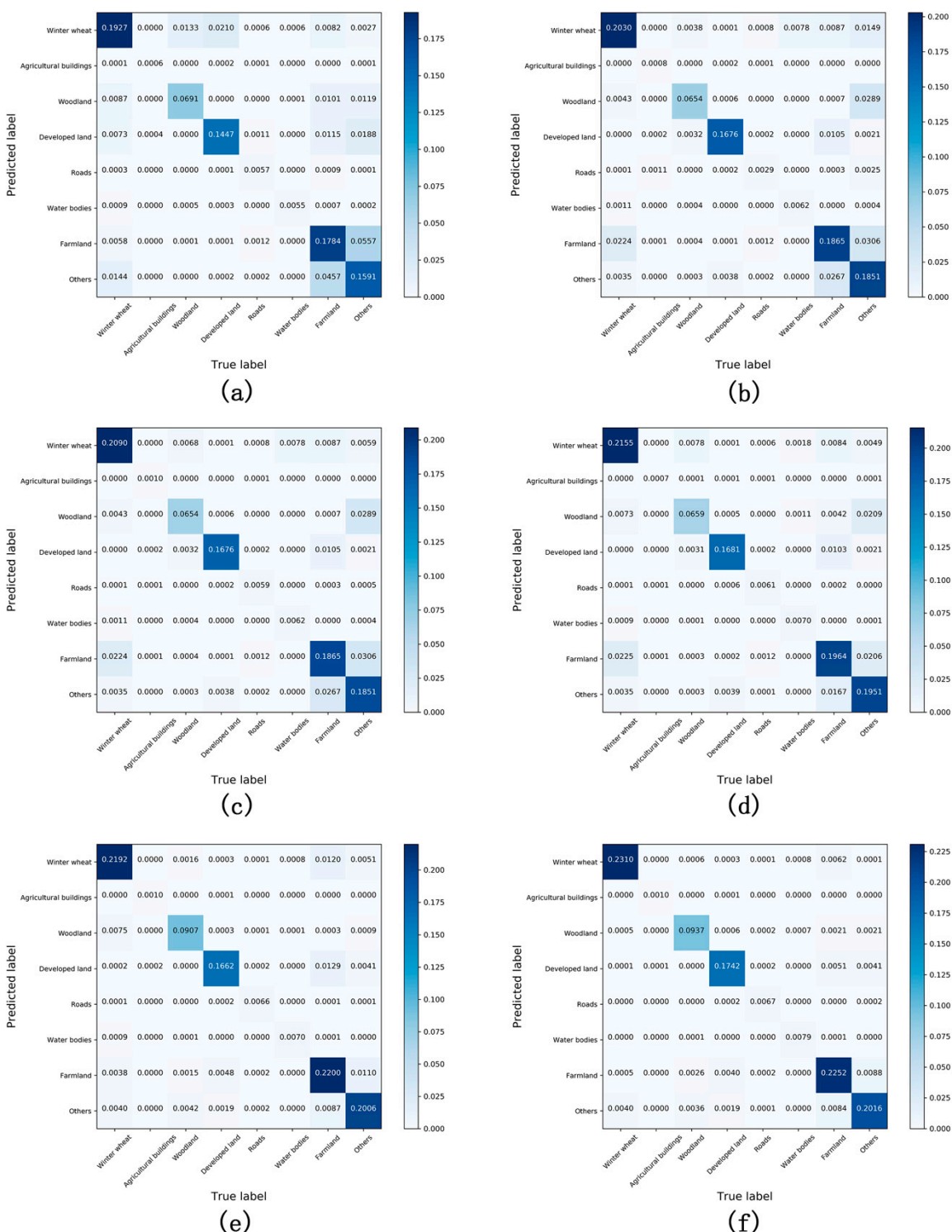

**Figure 8.** Confusion matrix for different model on the GF-6 images dataset: (**a**) SegNet, (**b**) U-Net, (**c**) RefineNet, (**d**) WFCNN-1, (**e**) WFCNN-2, and (**f**) WFCNN.

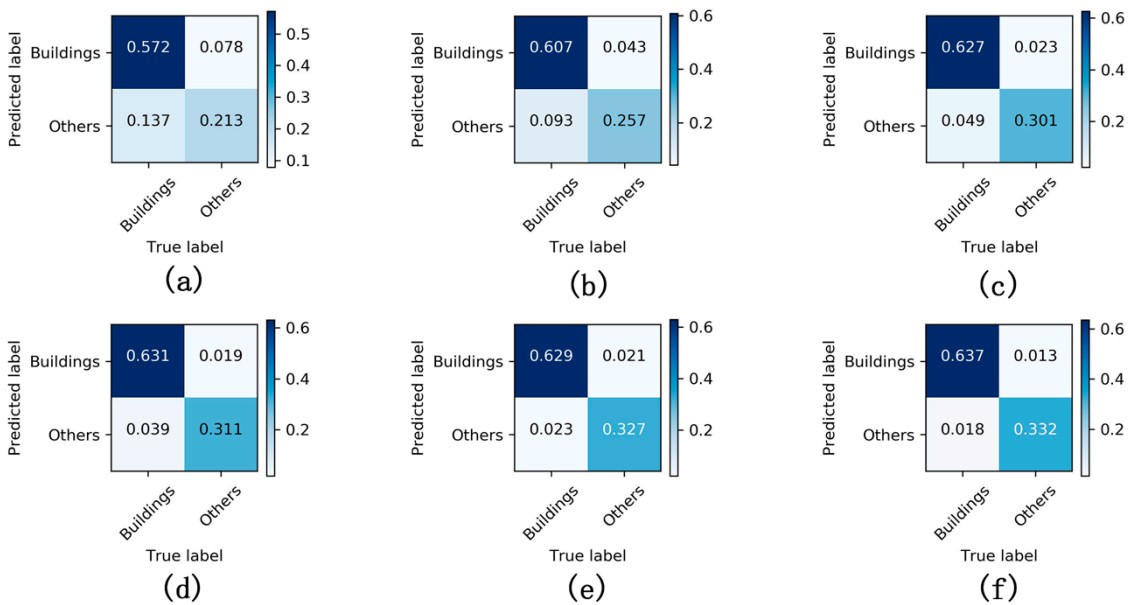

**Figure 9.** Confusion matrix for different model on the aerial image labeling dataset: (**a**) SegNet, (**b**) U-Net, (**c**) RefineNet, (**d**) WFCNN-1, (**e**) WFCNN-2, and (**f**) WFCNN.

For comparison, we use Table 5 to summarize the data given in Figure 8, and use Table 6 to summarize the data given in Figure 9.

**Table 5.** Comparison of model performance statistics on the GF-6 images dataset.

| Indicator | SegNet | U–Net | RefineNet | WFCNN-1 | WFCNN-2 | WFCNN |
|:---:|:---:|:---:|:---:|:---:|:---:|:---:|
| A | 75.58% | 81.75% | 82.67% | 85.48% | 91.13% | 94.13% |
| B | 24.42% | 18.25% | 17.33% | 14.52% | 8.87% | 5.87% |

A denotes the proportion of correctly classified pixels; B denotes the proportion of misclassified pixels.

**Table 6.** Comparison of model performance statistics on the aerial image labeling dataset.

| Indicator | SegNet | U–Net | RefineNet | WFCNN-1 | WFCNN-2 | WFCNN |
|:---:|:---:|:---:|:---:|:---:|:---:|:---:|
| A | 78.50% | 86.40% | 92.80% | 94.20% | 95.60% | 96.90% |
| B | 21.50% | 13.60% | 7.20% | 5.80% | 4.40% | 3.10% |

A denotes the proportion of correctly classified pixels; B denotes the proportion of misclassified pixels.

We used accuracy, precision, recall, F1-Score, intersection over union (IoU), and Kappa coefficient as indicators to further evaluate the segmentation results for each model (Tables 7 and 8). The F1-Score is defined as the harmonic mean of precision and recall. IoU is defined as the number of pixels labeled as the same class in both the prediction and the reference, divided by the number of pixels labeled as the class in the prediction or the reference.

The average accuracy of WFCNN was 18.48 % higher than SegNet, 11.44% higher than U-Net, 7.78% higher than RefineNet, 6.68% higher than WFCNN-1, and 2.15% higher than WFCNN-2.

**Table 7.** Comparison of model indicators on the GF-6 images dataset comparison.

| Indicator | SegNet | U-Net | RefineNet | WFCNN-1 | WFCNN-2 | WFCNN |
|---|---|---|---|---|---|---|
| Accuracy | 75.58% | 81.75% | 82.67% | 85.48% | 91.13% | 94.13% |
| Precision | 75.05% | 69.62% | 76.07% | 81.65% | 89.90% | 91.93% |
| Recall | 72.20% | 74.16% | 82.02% | 81.72% | 90.71% | 94.14% |
| F1-Score | 0.7101 | 0.7797 | 0.7904 | 0.8231 | 0.8906 | 0.9271 |
| IoU | 0.7360 | 0.7181 | 0.7893 | 0.8169 | 0.9031 | 0.9302 |
| Kappa coefficient | 0.5826 | 0.5652 | 0.6514 | 0.6905 | 0.8232 | 0.8693 |

**Table 8.** Comparison of model indicators on the aerial image labeling dataset.

| Indicator | SegNet | U-Net | RefineNet | WFCNN-1 | WFCNN-2 | WFCNN |
|---|---|---|---|---|---|---|
| Accuracy | 78.50% | 86.40% | 92.80% | 94.20% | 95.60% | 96.90% |
| Precision | 76.94% | 86.19% | 92.83% | 94.21% | 95.22% | 96.74% |
| Recall | 74.43% | 83.41% | 91.23% | 92.97% | 95.10% | 96.43% |
| F1-Score | 0.6117 | 0.7362 | 0.8522 | 0.8794 | 0.9077 | 0.9340 |
| IoU | 0.7566 | 0.8478 | 0.9202 | 0.9358 | 0.9516 | 0.9658 |
| Kappa coefficient | 0.6122 | 0.7355 | 0.8520 | 0.8793 | 0.9080 | 0.9341 |

## 5. Discussion

### 5.1. Effect of Features on Accuracy

At present, remote sensing image classification mainly relies on three feature types: spectral features mainly express information for individual pixels, textural features mainly express the spatial correlation between adjacent pixels, and semantic features mainly express the relationship between pixels in a specific region in a more abstract way.

CNNs can simultaneously extract these three feature types by reasonably organizing the convolution kernel. For example, the features extracted by using a $1 \times 1$ convolution kernel are equivalent to spectral features, those obtained by shallow convolutional layers can be regarded as textural features, and those obtained by convolutional layers of different depths can be regarded as semantic. Therefore, advanced semantic features can be considered as including all three feature types. However, in CNNs, advanced semantic features are obtained by deepening the network structure and expanding the receptive field. The spatial resolution of the image, the number of pixels covered by the object area, and other factors will affect the feature extraction results.

In a high-resolution and detailed camera image, an object tends to occupy a larger area, and it is advantageous to use advanced features. When SegNet only uses high-level semantic features to process an image with a high level of detail, a certain number of misclassified pixels occur at the edge, but few occur inside the object.

In the experiments of this study, the results of SegNet contained more misclassified pixels both at the edge and the inner area. Comparing objects of different sizes showed that smaller objects had more other types of objects adjacent to them and more pixels filled in. This is because the receptive field of high-level semantic features is generally large, and when the type distribution of objects in the receptive field is messy, the extracted feature values will deviate greatly from those of other pixels of the same kind, thus, affecting the accuracy of the segmentation results.

Unlike SegNet, the other models tested here combined low-level semantic features with high-level semantic features, with more accurate results. When segmenting remote sensing images with high spatial resolution, such as GF-6, the different levels of semantic features should be fused and used for classification, which is more reasonable than using advanced semantic features alone.

*5.2. Effect of Up-Sampling on Accuracy*

The purpose of up-sampling is to encrypt the rough high-level semantic feature graph to generate a feature vector for each pixel. In this study, WFCNN, U-Net, and RefineNet generated feature maps with the same size as compared to the input image through multi-step sampling. However, WFCNN used deconvolution to complete up-sampling, while RefineNet and U-Net used deconvolution to perform up-sampling first and chose linear interpolation for the final up-sampling. The segmentation results for these three models had few mis-segmented pixels within the object, but at the edges, WFCNN's results were significantly better. When RefineNet and U-Net processed camera images, the edges of the object were very fine. We believe that the reason for this phenomenon is that the structure of a remote sensing image is considerably different from that of a camera image, such that bilinear interpolation does not achieve good results when processing the former. Since the resolution of a camera image is generally high, when two objects are adjacent to each other, the pixel changes across the boundary are usually gentle and the difference between adjacent pixels is small, such that up-sampling using bilinear interpolation is more effective. In contrast, the pixel changes in a remote sensing image are usually sharp across the boundary between two objects and the difference between adjacent pixels is large, resulting in a poor bilinear interpolation effect. Unlike RefineNet and U-Net, WFCNN uses deconvolution for up-picking and correction and all parameters are obtained through learning samples, allowing the model to adapt to the unique characteristics of remote sensing images and achieve good results.

*5.3. Effect of Feature Fusion on Accuracy*

WFCNN, WFCNN-1, and WFCNN-2 all used feature fusion methods to generate feature vectors for each pixel. Since the convolution kernels used in different convolutional layer levels may be different, potentially causing differences in the feature map depths, it is necessary to adjust the latter before fusion. Choosing an appropriate fusion strategy can help improve the accuracy of the results.

WFCNN used a dimensional reduction strategy while WFCNN-1 used a dimensional increase. Although both can achieve consistent depth adjustments of the feature graph, our results showed that WFCNN performed better, indicating that dimensional reduction is more effective. Our analysis suggests that the effect of the classifier may be influenced by a high dimensionality. In a future study, we intend to test additional fusion strategies to further improve the accuracy of the segmentation results.

Like WFCNN, WFCNN-2 used dimensional reduction, but the former adjusted the characteristics of the participation fusion by using the weight layer beforehand, while the latter directly used dimensional reduction. Once the resulting feature map was fused, it was clear that the weight layer used by WFCNN improved the accuracy of the segmentation results.

## 6. Conclusions

This study tested a new approach to obtain high-precision land-use information from remote sensing images, using a new CNN-based model (WFCNN) to obtain multi-scale image features. Experimental comparisons with the currently used SegNet and RefineNet models, along with two variants on the original WFCNN model, demonstrated the advantages of this approach. This paper discusses the influence of classification feature selection, up-sampling method, and fusion strategy on segmentation accuracy, indicating that WFCNN can also be applied to other high spatial resolution remote sensing images.

Analyzing the data characteristics of high-resolution remote sensing images allowed the effects of spatial resolution on feature extraction to be fully considered, leading to the adoption of a hierarchical pooling strategy. In the initial stage of feature extraction, a large cell length step was used in WFCNN because the feature values were relatively scattered, allowing the pooling operation to accentuate the advantages of feature aggregation. In the subsequent stage, a smaller pooling step size was adopted

due to the large difference between the current resolution and the original image. The experimental results showed that this strategy was more conducive to generating features with good discrimination.

The data characteristics of adjacent regions within remote sensing images were fully considered to effectively combine low-level and high-level semantic features. The proposed WFCNN model uses a variable-weight fusion strategy that adjusts features using a weight adjustment layer to ensure their stability. This strategy is more conducive to extracting effective features than those adopted by other models.

Our method requires training images to be marked pixel-by-pixel, creating a large workload. In subsequent research, we intend to introduce a semi-supervised training method to reduce this requirement and make the model more applicable to real-world situations.

**Author Contributions:** Conceptualization, C.Z. and X.Y.; methodology, C.Z.; software, Y.C.; validation, S.G., X.Y. and F.L.; formal analysis, C.Z. and L.S.; investigation, A.K. and D.Z.; resources, F.L.; data curation, A.K.; writing—original draft preparation, C.Z.; writing—review and editing, C.Z.; visualization, S.G. and D.Z.; supervision, C.Z.; project administration, C.Z. and S.G.; funding acquisition, C.Z. All authors have read and agreed to the published version of the manuscript.

**Funding:** This research was funded by the Science Foundation of Shandong, grant numbers ZR2017MD018; the Key Research and Development Program of Ningxia, Grant numbers 2019BEH03008; the National Key R and D Program of China, grant number 2017YFA0603004; the Open Research Project of the Key Laboratory for Meteorological Disaster Monitoring, Early Warning and Risk Management of Characteristic Agriculture in Arid Regions, Grant numbers CAMF-201701 and CAMF-201803; the arid meteorological science research fund project by the Key Open Laboratory of Arid Climate Change and Disaster Reduction of CMA, Grant numbers IAM201801. The APC was funded by ZR2017MD018.

**Conflicts of Interest:** The authors declare no conflict of interest.

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
