# Peer review of "Improved Remote Sensing Image Classification Based on Multi-Scale Feature Fusion"

_remotesensing, doi:10.3390/rs12020213_

Round 1

Reviewer 1 Report

Thank you for your contribution. I appreciate your contribution and it has a good base.

My major comments are on the following points:

The word segmentation in the title is maybe not the most suitable as you measure and experiment improvement of classification results thus you should claim “Improved remote sensing image classification based on multi-scale feature fusion” Maybe some cross validation in the experiment could help the reader to have an idea of the robustness of the proposed approach I also suggest to add some geometric accuracy measures as your method seems to have some good visual results that could be validate with IoU measures for example. This of course requires highly geometric accurate reference datasets.

I have also some general comments/interrogations/suggestions for the revision:

L54: [13] taking L97: pest monitoring is very specific compared to object recognition. L139: and greenhouses. Bare lands refer to L216: the influence of this problem L256: could you justified the 8 number? Class number? Tables 5-9 are hard to read (global percentages) and hard to compare. I suggest that you give confusion matrices in pixel-counting or surfaces and add a table like table 10 to compare, per class, the F1-score per class results between the 8 classes Table 10: Kappa cannot be expressed in % as it could have value between -1 and 1. I suggest you add the F1-score also. L346-347: not clear, could you please rephrase?

Author Response

Response to reviewer's comments from Remote Sensing (remotesensing-664043)

Dear Reviewer:

We would like to thank you for the comments and suggestions. We have substantially revised the manuscript according to your good suggestions and detailed responses are provided below. All revised contents are in blue.

 (1) Thank you for your contribution. I appreciate your contribution and it has a good base.

Reply: We appreciate your support. We would like to thank you for the good comments and suggestions from you.

(2) The word segmentation in the title is maybe not the most suitable as you measure and experiment improvement of classification results thus you should claim “Improved remote sensing image classification based on multi-scale feature fusion”.

Reply: According to your good suggestions, we have revised the title. The revised title is as follow.

Improved remote sensing image classification based on multi-scale feature fusion

(3) Maybe some cross validation in the experiment could help the reader to have an idea of the robustness of the proposed approach.

Reply: According to your good suggestion, we redesigned the experimental plan and conducted comparative experiments. We have revised the Section 3.3, and added new content about using cross-validation in experiments. The added relevant contents are as follow.

We used cross-validation for comparative experiments. When using the GF-6 images dataset, 157 images had been randomly selected as test data every training session, and the other images had been used as training data until all images were tested once. When using the aerial image labeling dataset, 3000 images were used as test data every training session.

(4) I also suggest to add some geometric accuracy measures as your method seems to have some good visual results that could be validate with IoU measures for example. This of course requires highly geometric accurate reference datasets.

Reply: According to your good suggestion, we add intersection over union (IoU) as a new evaluation index. Each image in the two datasets we used in this study has a corresponding pixel-by-pixel label. We use these label data as the accurate reference data for calculating the IOU.

The revised relevant contents are as follow.

We used accuracy, precision, recall, F1-Score, intersection over union (IoU), and Kappa coefficient as indicators to further evaluate the segmentation results for each model (Table 7, Table 8). The F1-Score is defined as the harmonic mean of precision and recall. IoU is defined as the number of pixels labeled as the same class in both the prediction and the reference, divided by the number of pixels labeled as the class in the prediction or the reference.

The average accuracy of WFCNN was 18.48 % higher than SegNet, 11.44% higher than U-Net, 7.78% higher than RefineNet, 6.68% higher than WFCNN-1, and 2.15% higher than WFCNN-2.

Table 7. Comparison of model indicators on the GF-6 images dataset comparison.

Indicator

SegNet

U-Net

RefineNet

WFCNN-1

WFCNN-2

WFCNN

Accuracy

75.58%

81.75%

82.67%

85.48%

91.13%

94.13%

Precision

75.05%

69.62%

76.07%

81.65%

89.90%

91.93%

Recall

72.20%

74.16%

82.02%

81.72%

90.71%

94.14%

F1-Score

0.7101

0.7797

0.7904

0.8231

0.8906

0.9271

IoU

0.7360

0.7181

0.7893

0.8169

0.9031

0.9302

Kappa coefficient

0.5826

0.5652

0.6514

0.6905

0.8232

0.8693

Table 8. Comparison of model indicators on the aerial image labeling dataset.

Indicator

SegNet

U-Net

RefineNet

WFCNN-1

WFCNN-2

WFCNN

Accuracy

78.50%

86.40%

92.80%

94.20%

95.60%

96.90%

Precision

76.94%

86.19%

92.83%

94.21%

95.22%

96.74%

Recall

74.43%

83.41%

91.23%

92.97%

95.10%

96.43%

F1-Score

0.6117

0.7362

0.8522

0.8794

0.9077

0.9340

IoU

0.7566

0.8478

0.9202

0.9358

0.9516

0.9658

Kappa coefficient

0.6122

0.7355

0.8520

0.8793

0.9080

0.9341

(5) L54: [13] taking L97: pest monitoring is very specific compared to object recognition.

Reply: According to your good suggestion, we revised this sentence. The revised relevant contents are as follow.

As CNNs have outstanding advantages in feature extraction, they have been widely used in other fields, such as real-time traffic sign recognition [43], pedestrian recognition [44], apple target recognition [45], plant disease detection [46], and pest monitoring [46].

(6) L139: and greenhouses. Bare lands refer to

Reply: According to your good suggestion, we have corrected this mistake and revised relevant contents. The revised relevant contents are as follow.

There are eight main land-use types in the study area: developed land, water bodies, agricultural buildings, roads, farmland, winter wheat, woodland, and others. Developed land includes residential and factory areas, agricultural buildings refer to buildings in crop growing areas, and others refer to areas not used.

(7) L216: the influence of this problem

Reply: According to your good suggestion, we have corrected this mistake. The revised relevant contents are as follow.

When the pooling operation is carried out on images with a high level of detail or in the shallow convolutional structure, the influence of this problem is not obvious, but if this operation is carried out in the deep convolutional structure, the problem has a serious influence on feature extraction.

 (8) L256: could you justified the 8 number? Class number?

Reply: According to your good suggestion, we have corrected this mistake. The revised relevant contents are as follow.

WFCNN defines the loss function based on the cross entropy of the sample:

,

(1)

where m is the number of categories, p is the category probability vector with a length of m output by WFCNN, and q is the real probability distribution, generated according to the manual label. In q, the components are 0 except for that corresponding to the pixel’s category, where it is 1. In WFCNN, each pixel is regarded as an independent sample, allowing the loss function to be defined as:

,

(2)

where total represents the sample count.

(9) Tables 5-9 are hard to read (global percentages) and hard to compare. I suggest that you give confusion matrices in pixel-counting or surfaces and add a table like table 10 to compare, per class, the F1-score per class results between the 8 classes

Reply: According to your good suggestion, we reorganized the surface form of the confusion matrix. add a statistical table for every dataset. The revised relevant contents are as follow.

Figure 8 and Figure 9 present confusion matrices for the different models, which again demonstrate that WFCNN had the best segmentation results. Comparing Figure 8 and Figure 9, it can be found that the performance of each model on the aerial image labeling dataset is better than that on the GF-6 images dataset, indicating that the spatial resolution has a certain impact on the performance of the model.

Figure 8. Confusion matrix for different model on the GF-6 images dataset: (a) SegNet, (b) U-Net, (c) RefineNet, (d) WFCNN-1, (e) WFCNN-2, and (f) WFCNN.

Figure 9. Confusion matrix for different model on the aerial image labeling dataset: (a) SegNet, (b) U-Net, (c) RefineNet, (d) WFCNN-1, (e) WFCNN-2, and (f) WFCNN.

For comparison, we use Table 5 to summarize the data given in Figure 8, and use Table 6 to summarize the data given in Figure 9.

Table 5. Comparison of model performance statistics on the GF-6 images dataset.

Indicator

SegNet

U-Net

RefineNet

WFCNN-1

WFCNN-2

WFCNN

A

75.58%

81.75%

82.67%

85.48%

91.13%

94.13%

B

24.42%

18.25%

17.33%

14.52%

8.87%

5.87%

1A denotes the proportion of correctly classified pixels.

2B denotes the proportion of misclassified pixels.

Table 6. Comparison of model performance statistics on the aerial image labeling dataset.

Indicator

SegNet

U-Net

RefineNet

WFCNN-1

WFCNN-2

WFCNN

A

78.50%

86.40%

92.80%

94.20%

95.60%

96.90%

B

21.50%

13.60%

7.20%

5.80%

4.40%

3.10%

1A denotes the proportion of correctly classified pixels.

2B denotes the proportion of misclassified pixels.

We used accuracy, precision, recall, F1-Score, intersection over union (IoU), and Kappa coefficient as indicators to further evaluate the segmentation results for each model (Table 7, Table 8). The F1-Score is defined as the harmonic mean of precision and recall. IoU is defined as the number of pixels labeled as the same class in both the prediction and the reference, divided by the number of pixels labeled as the class in the prediction or the reference.

The average accuracy of WFCNN was 18.48 % higher than SegNet, 11.44% higher than U-Net, 7.78% higher than RefineNet, 6.68% higher than WFCNN-1, and 2.15% higher than WFCNN-2.

Table 7. Comparison of model indicators on the GF-6 images dataset comparison.

Indicator

SegNet

U-Net

RefineNet

WFCNN-1

WFCNN-2

WFCNN

Accuracy

75.58%

81.75%

82.67%

85.48%

91.13%

94.13%

Precision

75.05%

69.62%

76.07%

81.65%

89.90%

91.93%

Recall

72.20%

74.16%

82.02%

81.72%

90.71%

94.14%

F1-Score

0.7101

0.7797

0.7904

0.8231

0.8906

0.9271

IoU

0.7360

0.7181

0.7893

0.8169

0.9031

0.9302

Kappa coefficient

0.5826

0.5652

0.6514

0.6905

0.8232

0.8693

Table 8. Comparison of model indicators on the aerial image labeling dataset.

Indicator

SegNet

U-Net

RefineNet

WFCNN-1

WFCNN-2

WFCNN

Accuracy

78.50%

86.40%

92.80%

94.20%

95.60%

96.90%

Precision

76.94%

86.19%

92.83%

94.21%

95.22%

96.74%

Recall

74.43%

83.41%

91.23%

92.97%

95.10%

96.43%

F1-Score

0.6117

0.7362

0.8522

0.8794

0.9077

0.9340

IoU

0.7566

0.8478

0.9202

0.9358

0.9516

0.9658

Kappa coefficient

0.6122

0.7355

0.8520

0.8793

0.9080

0.9341

(11) Table 10: Kappa cannot be expressed in % as it could have value between -1 and 1. I suggest you add the F1-score also.

Reply: According to your good suggestion, we have corrected this mistake. The revised relevant contents are as follow.

Table 7. Comparison of model indicators on the GF-6 images dataset comparison.

Indicator

SegNet

U-Net

RefineNet

WFCNN-1

WFCNN-2

WFCNN

Accuracy

75.58%

81.75%

82.67%

85.48%

91.13%

94.13%

Precision

75.05%

69.62%

76.07%

81.65%

89.90%

91.93%

Recall

72.20%

74.16%

82.02%

81.72%

90.71%

94.14%

F1-Score

0.7101

0.7797

0.7904

0.8231

0.8906

0.9271

IoU

0.7360

0.7181

0.7893

0.8169

0.9031

0.9302

Kappa coefficient

0.5826

0.5652

0.6514

0.6905

0.8232

0.8693

Table 8. Comparison of model indicators on the aerial image labeling dataset.

Indicator

SegNet

U-Net

RefineNet

WFCNN-1

WFCNN-2

WFCNN

Accuracy

78.50%

86.40%

92.80%

94.20%

95.60%

96.90%

Precision

76.94%

86.19%

92.83%

94.21%

95.22%

96.74%

Recall

74.43%

83.41%

91.23%

92.97%

95.10%

96.43%

F1-Score

0.6117

0.7362

0.8522

0.8794

0.9077

0.9340

IoU

0.7566

0.8478

0.9202

0.9358

0.9516

0.9658

Kappa coefficient

0.6122

0.7355

0.8520

0.8793

0.9080

0.9341

(12) L346-347: not clear, could you please rephrase?

Reply: According to your good suggestion, we revised this sentence. The revised relevant contents are as follow.

In the experiments of this study, the results of SegNet contained more misclassified pixels both at the edge and the inner area.

Reviewer 2 Report

The manuscript targets a very interesting topic of segmentation of remotely sensed images with particular focus on multi-scale features fusion. Overall, the paper is interesting, however, there are several limitations including

The language of the paper needs to be improved. There are some discontinuities in text in different sections. There are also some grammatical mistakes. Moreover, in most of the sections, unnecessary details are provided, which could be cut down to make it more precise. Some recent references are missing, such as Ahmad, Kashif, and Nicola Conci. "How Deep Features Have Improved Event Recognition in Multimedia: A Survey." Krylov, Vladimir, Eamonn Kenny, and Rozenn Dahyot. "Automatic discovery and geotagging of objects from street view imagery." Remote Sensing 10.5 (2018): 661. Helber, Patrick, et al. "Eurosat: A novel dataset and deep learning benchmark for land use and land cover classification." IEEE Journal of Selected Topics in Applied Earth Observations and Remote Sensing 12.7 (2019): 2217-2226.

 ACM Transactions on Multimedia Computing, Communications, and Applications (TOMM) 15.2 (2019): 39. The contributions of the paper are not clear. The authors should explicitly mention the main contributions of the paper in the introduction. A bullet wise description of the main contribution would make it clearer. The authors provide comparisons against two SoA models on the newly collected dataset, however, it would be interesting to compare the performance of benchmark dataset.

Author Response

Response to reviewer's comments from Remote Sensing (remotesensing-664043)

Dear Reviewer:

We would like to thank you for the comments and suggestions. We have substantially revised the manuscript according to your good suggestions and detailed responses are provided below. All revised contents are in blue.

(1) The manuscript targets a very interesting topic of segmentation of remotely sensed images with particular focus on multi-scale features fusion. Overall, the paper is interesting, however, there are several limitations including.

Reply: We appreciate your support. We would like to thank you for the good comments and suggestions from you.

(2) The language of the paper needs to be improved. There are some discontinuities in text in different sections. There are also some grammatical mistakes. Moreover, in most of the sections, unnecessary details are provided, which could be cut down to make it more precise.

Reply: According to your good suggestion, we employed a professional English editor to check and edit the full text in English, and corrected grammatical errors in the original text.

(3) Some recent references are missing, such as Ahmad, Kashif, and Nicola Conci. "How Deep Features Have Improved Event Recognition in Multimedia: A Survey." ACM Transactions on Multimedia Computing, Communications, and Applications (TOMM) 15.2 (2019): 39. Krylov, Vladimir, Eamonn Kenny, and Rozenn Dahyot. "Automatic discovery and geotagging of objects from street view imagery." Remote Sensing 10.5 (2018): 661. Helber, Patrick, et al. "Eurosat: A novel dataset and deep learning benchmark for land use and land cover classification." IEEE Journal of Selected Topics in Applied Earth Observations and Remote Sensing 12.7 (2019): 2217-2226.

Reply: According to your good suggestion, we had added these recent references. The revised relevant contents are as follow.

As CNNs have outstanding advantages in feature extraction, they have been widely used in other fields, such as real-time traffic sign recognition [43], pedestrian recognition [44], apple target recognition [45], plant disease detection [46], and pest monitoring [46]. Researchers have established a method to extract coordinate information of an object from street view imagery [48] using CNNs.

Compared to camera images, remote sensing images have fewer details and more mixed pixels. When using a CNN to extract information from remote sensing images, the influence of the convolution structure on feature extraction must be considered [49]. Existing CNN structures are mainly designed for camera images with a high level of detail; therefore, a more ideal result can be obtained through adjustments that consider the specific characteristics of remote sensing images [50,51]. Researchers have proposed a series of such adjustments for applying CNNs to remote sensing image processing [3] and some classic CNNs have been widely applied in this field [52, 53]. Based on the characteristic analysis of target objects and specific remote sensing imagery, researchers have established a series of CNNs such as two-branch CNN [49], WFS-NET [54], patch-based CNN [55], and hybrid MLP-CNN [56]. Researchers have also used remote sensing images to create many benchmark datasets, such as EuroSAT [57] or the Inria Aerial Image dataset [58], to test the performance of CNNs.

Krylov, V. A.; Kenny, E.; Dahyot, R. Automatic discovery and geotagging of objects from street view imagery. Remote Sens. 2018, 661, doi:10.3390/rs10050661. Ahmad, K.; Conci, N. How Deep Features Have Improved Event Recognition in Multimedia: A Survey. ACM Trans. Multimedia Comput. Commun. Appl. 2019, 39, doi: /10.1145/3306240.

Helber, P.; Bischke, B.; Dengel, A.; Borth, D. Eurosat: A novel dataset and deep learning benchmark for land use and land cover classification. IEEE Journal of Selected Topics in Applied Earth Observations and Remote Sensing. 2019,12,2217-2226, doi:10.1109/JSTARS.2019.2918242.

(4) The contributions of the paper are not clear. The authors should explicitly mention the main contributions of the paper in the introduction. A bullet wise description of the main contribution would make it clearer.

Reply: According to your good suggestion, we add contents in the introduction to highlighting the contributions. The revised relevant contents are as follow.

In this study, we established a CNN structure based on variable weight fusion, the Weight Feature Value Convolutional Neural Network (WFCNN), and experimentally assessed its performance. The main contributions of this work are as follows.

Based on the analysis of the data characteristics of remote sensing images, fully considering the impact of image spatial resolution on feature extraction, we establish a suitable network convolution structure; The proposed approach can effectively fuse low-level semantic features with high-level semantic features and fully considers the data characteristics of adjacent areas around objects on remote sensing images. Compared with the strategies adopted by other models, our approach is more conducive to effective features.

 (5) The authors provide comparisons against two SoA models on the newly collected dataset, however, it would be interesting to compare the performance of benchmark dataset.

Reply: According to your good suggestion, we add the aerial image labeling dataset, a benchmark dataset, downloaded from https://project.inria.fr/aerialimagelabeling/, to compare the performance of different. Firstly, we reorganized Section 2, added Section 2.2 to describe the aerial image labeling dataset. Secondly, we redesigned the experimental plan and conducted comparative experiments, revised Section 3.3. Thirdly, we revised Section 4. Finally, we added the reference of the aerial image labeling dataset. The revised relevant contents are as follow.

2.2 The Aerial Image Labeling Dataset

The aerial image labeling dataset was downloaded from https://project.inria.fr/aerialimagelabeling/ [58]. The images cover dissimilar urban settlements, ranging from densely populated areas (e.g., the financial district of San Francisco) to alpine towns (e.g., Lienz in the Austrian Tyrol). The dataset features were as follows:

Coverage of 810 km² Aerial orthorectified color imagery with a spatial resolution of 0.3 m Ground truth data for two semantic classes: building and no building

The original aerial image labeling dataset contained 180 color image tiles, 5000×5000 pixels in size, which we cropped into small image patches, each with a size of 500×500 pixels. an image-label pair example is provided in Figure 4.

Figure 4. Example of image-label pair: (a) original image and (b) classified image.

3.3. Experimental Setup

We used the SegNet, U-Net, and RefineNet models for comparison with WFCNN, as their structures and working principles are similar, providing the best test for WFCNN. The models were set up based on previous research, with SegNet containing 13 convolutional layers [39], U-Net containing 10 convolutional layers [42], and RefineNet containing 101 convolutional layers [41]. We implemented WFCNN based on the TensorFlow Framework, using the Python language. To better assess its performance, we also tested two variants, termed WFCNN-1 and WFCNN-2 (Table 4).

Table 4. Models used in the comparative experiment.

Name

Description

WFCNN

SegNet

Similar to WFCNN, the classifier used only high-level semantic features.

U-Net

Similar to WFCNN, the classifier used fused features.

RefineNet

Similar to WFCNN, the linear model was also adopted for feature fusion, but the parameters were all fixed as 1.

WFCNN-1

The decoding unit was modified to use an adjustment strategy for the feature map depth ascending scale. The length of the feature vector generated by the decoder was 512 for each pixel.

WFCNN-2

The decoding unit was modified to remove the weight layer.

All experiments were conducted on a graphics workstation with a 12 GB NVIDIA graphics card and the Linux Ubuntu 16.04 operating system, using the data set defined in section 2.

To increase the number and diversity of samples, each image in the training data set was processed using color adjustment, horizontal flip, and vertical flip steps. The color adjustment factors included brightness, saturation, hue, and contrast, and each image was processed 10 times. These enhanced images were only used as training data.

We used cross-validation for comparative experiments. When using the GF-6 images dataset, 157 images had been randomly selected as test data every training session, and the other images had been used as training data until all images were tested once. When using the aerial image labeling dataset, 3000 images were used as test data every training session.

Results

Overall, 10 results tested on the GF-6 images dataset were randomly selected from all models (Figure 6), and 10 results were tested on the aerial image labeling dataset (Figure 7). As can be seen from Figure 6 and 7, WFCNN performed best in all cases.

SegNet exhibited most errors and the distribution was relatively scattered, with more misclassified pixels in both the edge and inner areas. This shows that it is more reasonable to combine high semantic features with low semantic features than to use only high semantic features.

The number of misclassified pixels produced by all three variants of WFCNN was lower for RefineNet and U-net. The excellent performance of RefineNet and U-Net in the camera image indicates that the network structure should be determined in accordance with the spatial resolution of the image.

WFCNN performed better than WFCNN-1. This excellent performance indicates that the feature vector dimension was too high and is not conducive to improving the accuracy of the classifier. The result that WFCNN performed better than WFCNN-2 shows that the weight layer played a role.

By comparing the performance of U-Net, RefineNet, WFCNN-1, WFCNN-2, WFCNN, it indicates different feature fusion methods differ in their contributions to improving accuracy, so it is necessary to choose an appropriate feature fusion method for a given situation.

Figure 6. Comparative experimental results for 10 selected images from the GF-6 images dataset: (a) original GF-6 image, (b) manual classification, (c) SegNet, (d) U-Net, (e) RefineNet, (f) WFCNN-1, (g) WFCNN-2, and (h) WFCNN.

Figure 7. Comparative experimental results for 10 selected images from the aerial image labeling dataset: (a) original image, (b) manual classification, (c) SegNet, (d) U-Net, (e) RefineNet, (f) WFCNN-1, (g) WFCNN-2, and (h) WFCNN.

Figure 8 and Figure 9 present confusion matrices for the different models, which again demonstrate that WFCNN had the best segmentation results. Comparing Figure 8 and Figure 9, it can be found that the performance of each model on the aerial image labeling dataset is better than that on the GF-6 images dataset, indicating that the spatial resolution has a certain impact on the performance of the model.

Figure 8. Confusion matrix for different model on the GF-6 images dataset: (a) SegNet, (b) U-Net, (c) RefineNet, (d) WFCNN-1, (e) WFCNN-2, and (f) WFCNN.

Figure 9. Confusion matrix for different model on the aerial image labeling dataset: (a) SegNet, (b) U-Net, (c) RefineNet, (d) WFCNN-1, (e) WFCNN-2, and (f) WFCNN.

For comparison, we use Table 5 to summarize the data given in Figure 8, and use Table 6 to summarize the data given in Figure 9.

Table 5. Comparison of model performance statistics on the GF-6 images dataset.

Indicator

SegNet

U-Net

RefineNet

WFCNN-1

WFCNN-2

WFCNN

A

75.58%

81.75%

82.67%

85.48%

91.13%

94.13%

B

24.42%

18.25%

17.33%

14.52%

8.87%

5.87%

1A denotes the proportion of correctly classified pixels.

2B denotes the proportion of misclassified pixels.

Table 6. Comparison of model performance statistics on the aerial image labeling dataset.

Indicator

SegNet

U-Net

RefineNet

WFCNN-1

WFCNN-2

WFCNN

A

78.50%

86.40%

92.80%

94.20%

95.60%

96.90%

B

21.50%

13.60%

7.20%

5.80%

4.40%

3.10%

1A denotes the proportion of correctly classified pixels.

2B denotes the proportion of misclassified pixels.

We used accuracy, precision, recall, F1-Score, intersection over union (IoU), and Kappa coefficient as indicators to further evaluate the segmentation results for each model (Table 7, Table 8). The F1-Score is defined as the harmonic mean of precision and recall. IoU is defined as the number of pixels labeled as the same class in both the prediction and the reference, divided by the number of pixels labeled as the class in the prediction or the reference.

The average accuracy of WFCNN was 18.48 % higher than SegNet, 11.44% higher than U-Net, 7.78% higher than RefineNet, 6.68% higher than WFCNN-1, and 2.15% higher than WFCNN-2.

Table 7. Comparison of model indicators on the GF-6 images dataset comparison.

Indicator

SegNet

U-Net

RefineNet

WFCNN-1

WFCNN-2

WFCNN

Accuracy

75.58%

81.75%

82.67%

85.48%

91.13%

94.13%

Precision

75.05%

69.62%

76.07%

81.65%

89.90%

91.93%

Recall

72.20%

74.16%

82.02%

81.72%

90.71%

94.14%

F1-Score

0.7101

0.7797

0.7904

0.8231

0.8906

0.9271

IoU

0.7360

0.7181

0.7893

0.8169

0.9031

0.9302

Kappa coefficient

0.5826

0.5652

0.6514

0.6905

0.8232

0.8693

Table 8. Comparison of model indicators on the aerial image labeling dataset.

Indicator

SegNet

U-Net

RefineNet

WFCNN-1

WFCNN-2

WFCNN

Accuracy

78.50%

86.40%

92.80%

94.20%

95.60%

96.90%

Precision

76.94%

86.19%

92.83%

94.21%

95.22%

96.74%

Recall

74.43%

83.41%

91.23%

92.97%

95.10%

96.43%

F1-Score

0.6117

0.7362

0.8522

0.8794

0.9077

0.9340

IoU

0.7566

0.8478

0.9202

0.9358

0.9516

0.9658

Kappa coefficient

0.6122

0.7355

0.8520

0.8793

0.9080

0.9341

the added reference is as follow.

Maggiori, E.; Tarabalka, Y.; Charpiat, G.; Alliez. P. Can Semantic Labeling Methods Generalize to Any City? The Inria Aerial Image Labeling Benchmark. IEEE International Symposium on Geoscience and Remote Sensing (IGARSS), 2017,6, hal-01468452.

Reviewer 3 Report

The study "Improved remote sensing image segmentation based
3 on multi-scale feature fusion" presents an neural network based approach to segment different land uses in remote sensing images. I like, that the authors explained their concept really clear and the results seem sound.

I have one major point that is missing in this study. U-Nets are the state of the art segmentation network and they are completely missing in this study. I would suggest to add this to the comparison in order to achieve an holisitic comparison to existing methods.

Author Response

Response to reviewer's comments from Remote Sensing (remotesensing-664043)

Dear Reviewer:

We would like to thank you for the comments and suggestions. We have substantially revised the manuscript according to your good suggestions and detailed responses are provided below. All revised contents are in blue.

 (1) The study "Improved remote sensing image segmentation based on multi-scale feature fusion" presents an neural network based approach to segment different land uses in remote sensing images. I like, that the authors explained their concept really clear and the results seem sound.

Reply: We appreciate your support. We would like to thank you for the good comments and suggestions from you.

(2) I have one major point that is missing in this study. U-Nets are the state of the art segmentation network and they are completely missing in this study. I would suggest to add this to the comparison in order to achieve an holisitic comparison to existing methods.

Reply: According to your good suggestion, we revised relevant contents. Firstly, we redesigned the experimental plan and conducted comparative experiments, revised Section 3.3., Secondly, we revised Section 4. Finally, we revised Section 5.2. The revised relevant contents are as follow.

3.3. Experimental Setup

We used the SegNet, U-Net, and RefineNet models for comparison with WFCNN, as their structures and working principles are similar, providing the best test for WFCNN. The models were set up based on previous research, with SegNet containing 13 convolutional layers [39], U-Net containing 10 convolutional layers [42], and RefineNet containing 101 convolutional layers [41]. We implemented WFCNN based on the TensorFlow Framework, using the Python language. To better assess its performance, we also tested two variants, termed WFCNN-1 and WFCNN-2 (Table 4).

Table 4. Models used in the comparative experiment.

Name

Description

WFCNN

SegNet

Similar to WFCNN, the classifier used only high-level semantic features.

U-Net

Similar to WFCNN, the classifier used fused features.

RefineNet

Similar to WFCNN, the linear model was also adopted for feature fusion, but the parameters were all fixed as 1.

WFCNN-1

The decoding unit was modified to use an adjustment strategy for the feature map depth ascending scale. The length of the feature vector generated by the decoder was 512 for each pixel.

WFCNN-2

The decoding unit was modified to remove the weight layer.

All experiments were conducted on a graphics workstation with a 12 GB NVIDIA graphics card and the Linux Ubuntu 16.04 operating system, using the data set defined in section 2.

To increase the number and diversity of samples, each image in the training data set was processed using color adjustment, horizontal flip, and vertical flip steps. The color adjustment factors included brightness, saturation, hue, and contrast, and each image was processed 10 times. These enhanced images were only used as training data.

We used cross-validation for comparative experiments. When using the GF-6 images dataset, 157 images had been randomly selected as test data every training session, and the other images had been used as training data until all images were tested once. When using the aerial image labeling dataset, 3000 images were used as test data every training session.

Results

Overall, 10 results tested on the GF-6 images dataset were randomly selected from all models (Figure 6), and 10 results were tested on the aerial image labeling dataset (Figure 7). As can be seen from Figure 6 and 7, WFCNN performed best in all cases.

SegNet exhibited most errors and the distribution was relatively scattered, with more misclassified pixels in both the edge and inner areas. This shows that it is more reasonable to combine high semantic features with low semantic features than to use only high semantic features.

The number of misclassified pixels produced by all three variants of WFCNN was lower for RefineNet and U-net. The excellent performance of RefineNet and U-Net in the camera image indicates that the network structure should be determined in accordance with the spatial resolution of the image.

WFCNN performed better than WFCNN-1. This excellent performance indicates that the feature vector dimension was too high and is not conducive to improving the accuracy of the classifier. The result that WFCNN performed better than WFCNN-2 shows that the weight layer played a role.

By comparing the performance of U-Net, RefineNet, WFCNN-1, WFCNN-2, WFCNN, it indicates different feature fusion methods differ in their contributions to improving accuracy, so it is necessary to choose an appropriate feature fusion method for a given situation.

Figure 6. Comparative experimental results for 10 selected images from the GF-6 images dataset: (a) original GF-6 image, (b) manual classification, (c) SegNet, (d) U-Net, (e) RefineNet, (f) WFCNN-1, (g) WFCNN-2, and (h) WFCNN.

Figure 7. Comparative experimental results for 10 selected images from the aerial image labeling dataset: (a) original image, (b) manual classification, (c) SegNet, (d) U-Net, (e) RefineNet, (f) WFCNN-1, (g) WFCNN-2, and (h) WFCNN.

Figure 8 and Figure 9 present confusion matrices for the different models, which again demonstrate that WFCNN had the best segmentation results. Comparing Figure 8 and Figure 9, it can be found that the performance of each model on the aerial image labeling dataset is better than that on the GF-6 images dataset, indicating that the spatial resolution has a certain impact on the performance of the model.

Figure 8. Confusion matrix for different model on the GF-6 images dataset: (a) SegNet, (b) U-Net, (c) RefineNet, (d) WFCNN-1, (e) WFCNN-2, and (f) WFCNN.

Figure 9. Confusion matrix for different model on the aerial image labeling dataset: (a) SegNet, (b) U-Net, (c) RefineNet, (d) WFCNN-1, (e) WFCNN-2, and (f) WFCNN.

For comparison, we use Table 5 to summarize the data given in Figure 8, and use Table 6 to summarize the data given in Figure 9.

Table 5. Comparison of model performance statistics on the GF-6 images dataset.

Indicator

SegNet

U-Net

RefineNet

WFCNN-1

WFCNN-2

WFCNN

A

75.58%

81.75%

82.67%

85.48%

91.13%

94.13%

B

24.42%

18.25%

17.33%

14.52%

8.87%

5.87%

1A denotes the proportion of correctly classified pixels.

2B denotes the proportion of misclassified pixels.

Table 6. Comparison of model performance statistics on the aerial image labeling dataset.

Indicator

SegNet

U-Net

RefineNet

WFCNN-1

WFCNN-2

WFCNN

A

78.50%

86.40%

92.80%

94.20%

95.60%

96.90%

B

21.50%

13.60%

7.20%

5.80%

4.40%

3.10%

1A denotes the proportion of correctly classified pixels.

2B denotes the proportion of misclassified pixels.

We used accuracy, precision, recall, F1-Score, intersection over union (IoU), and Kappa coefficient as indicators to further evaluate the segmentation results for each model (Table 7, Table 8). The F1-Score is defined as the harmonic mean of precision and recall. IoU is defined as the number of pixels labeled as the same class in both the prediction and the reference, divided by the number of pixels labeled as the class in the prediction or the reference.

The average accuracy of WFCNN was 18.48 % higher than SegNet, 11.44% higher than U-Net, 7.78% higher than RefineNet, 6.68% higher than WFCNN-1, and 2.15% higher than WFCNN-2.

Table 7. Comparison of model indicators on the GF-6 images dataset comparison.

Indicator

SegNet

U-Net

RefineNet

WFCNN-1

WFCNN-2

WFCNN

Accuracy

75.58%

81.75%

82.67%

85.48%

91.13%

94.13%

Precision

75.05%

69.62%

76.07%

81.65%

89.90%

91.93%

Recall

72.20%

74.16%

82.02%

81.72%

90.71%

94.14%

F1-Score

0.7101

0.7797

0.7904

0.8231

0.8906

0.9271

IoU

0.7360

0.7181

0.7893

0.8169

0.9031

0.9302

Kappa coefficient

0.5826

0.5652

0.6514

0.6905

0.8232

0.8693

Table 8. Comparison of model indicators on the aerial image labeling dataset.

Indicator

SegNet

U-Net

RefineNet

WFCNN-1

WFCNN-2

WFCNN

Accuracy

78.50%

86.40%

92.80%

94.20%

95.60%

96.90%

Precision

76.94%

86.19%

92.83%

94.21%

95.22%

96.74%

Recall

74.43%

83.41%

91.23%

92.97%

95.10%

96.43%

F1-Score

0.6117

0.7362

0.8522

0.8794

0.9077

0.9340

IoU

0.7566

0.8478

0.9202

0.9358

0.9516

0.9658

Kappa coefficient

0.6122

0.7355

0.8520

0.8793

0.9080

0.9341

5.2. Effect of Up-sampling on Accuracy

The purpose of up-sampling is to encrypt the rough high-level semantic feature graph to generate a feature vector for each pixel. In this study, WFCNN, U-Net, and RefineNet generated feature maps with the same size as compared to the input image through multi-step sampling. However, WFCNN used deconvolution to complete up-sampling, while RefineNet and U-Net used deconvolution to perform up-sampling first and chose linear interpolation for the final up-sampling. The segmentation results for these three models had few mis-segmented pixels within the object, but at the edges, WFCNN’s results were significantly better. When RefineNet and U-Net processed camera images, the edges of the object were very fine. We believe that the reason for this phenomenon is that the structure of a remote sensing image is considerably different from that of a camera image, such that bilinear interpolation does not achieve good results when processing the former. Since the resolution of a camera image is generally high, when two objects are adjacent to each other, the pixel changes across the boundary are usually gentle and the difference between adjacent pixels is small, such that up-sampling using bilinear interpolation is more effective. In contrast, the pixel changes in a remote sensing image are usually sharp across the boundary between two objects and the difference between adjacent pixels is large, resulting in a poor bilinear interpolation effect. Unlike RefineNet and U-Net, WFCNN uses deconvolution for up-picking and correction and all parameters are obtained through learning samples, allowing the model to adapt to the unique characteristics of remote sensing images and achieve good results.

Round 2

Reviewer 2 Report

My concerns have been addressed; i would go for acceptance